# Speech-in-noise performance in objectively determined cochlear implant maps, including the effect of cognitive state

**Jessica M. Yiannos[1], Christofer W. Bester[1,2], Christopher Zhao[2], Barbara Gell[3], Dona M. P. Jayakody[1,4,5,6]***

**1** Ear Science Institute Australia, Subiaco, Western Australia, Australia, **2** School of Human Sciences, University of Western Australia, Crawley, Western Australia, Australia, **3** MED-EL Australasia, Alexandria, New South Wales, Australia, **4** Centre for Ear Sciences, Medical School, University of Western Australia, Crawley, Western Australia, Australia, **5** Western Australian Centre for Health & Ageing, University of Western Australia, Crawley, Western Australia, Australia, **6** School of Allied Health, Curtin University, Bentley, Western Australia, Australia

* dona.jayakody@uwa.edu.au

**Data Availability Statement:** Data access is restricted due to a small subject population with correlations between identifiable information (e.g., audiological outcomes, cognitive outcomes and

## Abstract

### Objective

Cochlear Implant (CI) programming based on subjective psychophysical fine-tuning of loudness scaling involves active participation and cognitive skills and thus may not be appropriate for difficult-to-condition populations. The electrically evoked stapedial reflex threshold (eSRT) is an objective measure that is suggested to provide clinical benefit to CI programming. This study aimed to compare speech reception outcomes between subjective and eSRT objectively determined CI maps for adult MED-EL recipients. The effect of cognitive skills on these skills was further assessed.

### Methods

Twenty-seven post lingually hearing-impaired MED-EL CI recipients were recruited, 6 with mild cognitive impairment (MCI– 4 male, 79 years ± 5), 21 with normal cognitive function (5 male, 63 years ± 12). Two MAPs were generated: a subjective MAP and an objective MAP in which eSRTs determined maximum comfortable levels (M-Levels). Participants were randomly divided into two groups. Group A trialled the objective MAP for two weeks before returning for outcome assessment. During the following two weeks, Group A trialled the subjective MAP before returning for outcome assessment. Group B trialled MAPs in reverse. Outcome measures included the Hearing Implant Sound Quality Index (HISQUI), Consonant-Nucleus-Consonant (CNC) word test, and Bamford-Kowal-Bench Speech-in-Noise (BKB-SIN) test.

### Results

eSRT based MAPS were obtained in 23 of the participants. A strong relationship was demonstrated between global charge between eSRT-based and psychophysical-based M-Levels (r = 0.89, *p* < .001). The Montreal Cognitive Assessment for the Hearing Impaired

implant details). More than 3 indirect identifiers are required to complete analyses. Data are available upon request from the data custodian, Prof. Rob Eikelboom, via email (rob.eikelboom@earscience.org.au) for researchers who meet the criteria for access to confidential data.

**Funding:** The author(s) received no specific funding for this work.

**Competing interests:** It is disclosed that one of the authors is a clinical specialist working for MED-EL Australasia. This does not alter our adherence to PLOS ONE policies on sharing data and materials.

(MoCA-HI) testing identified 6 CI recipients with MCI (MoCA-HI total score ≤23). The MCI group was older (63, 79 years), but were not otherwise different in sex, duration of hearing loss or duration of CI use. For all patients, no significant differences were found for sound quality or speech in quiet scores between eSRT-based and psychophysical-based MAPs. However, psychophysically determined MAPs showed significantly better speech-in-noise reception (6.74 vs 8.20-dB SNR, $p$ = .34). MoCA-HI scores showed a significant, moderate negative correlation with BKB SIN for both MAP approaches (Kendall's Tau B, $p$ = .015 and $p$ = .008), with no effect on the difference between MAP approaches.

## Conclusion

Results indicate eSRT-based methods provide poorer outcomes than psychophysical-based method. While speech-in-noise reception is correlated with MoCA-HI score, this affected both behaviourally and objectively determined MAPs. The results suggest fair confidence in the eSRT-based method as a guide for setting M-Level for difficult-to-condition CI populations in simple listening conditions.

## Introduction

Cochlear implants (CI) are remarkably effective at generating the perception of sound using electrical stimulation. Each CI recipient has a personalized MAP, a set of parameters that defines the functional range of that participant's implant. The MAP includes, for each electrode, both the minimum stimulation that generates a percept (threshold, or T-level), as well as the maximum comfortable stimulation (C- or M-level). The parameters of the MAP contribute substantially to each patient's speech performance [1], in particular inappropriately low M-Levels lead to reduced speech performance and high M-Levels can lead to painful overstimulation [2, 3]. CI maps are known to be affected by intracochlear factors particularly the position of the electrode array relative to the spiral ganglion, and the density of residual spiral ganglion cells [4].

The typical CI programming procedure is through subjective fine-tuning, whereby parameter adjustments are made using psychophysical loudness scaling procedures [5, 6]. This subjective mapping method requires active user participation, requiring attention, patience, cognitive and language skills necessary to complete complex tasks with reliability and repeatability [7, 8]. Therefore, this programming method may not be suitable for CI population subgroups, including children, individuals with pronounced deficits with speech and language or individuals with cognitive impairments. Reliable, objective measures are preferred for these sub-groups who may not provide clear subjective loudness judgements to establish an appropriate dynamic range for speech perception [7, 9].

Objective mapping procedures are proposed to provide valuable information for speech processor programming [10], i.e., neural, physiological responses elicited via electrical stimuli generated from CI electrodes [11]. Initial investigations used the electrically-evoked compound actional potential (eCAP) or electrically-evoked auditory brainstem response (eABR), however prior work has shown weak correlations with behaviourally-determined MAPs, and a tendency to overestimate comfort levels, resulting in over-stimulation in 28% of cases [12, 13]. More recently, the electrically-evoked stapedial reflex (eSRT) has been investigated as a more sensitive alternative [14–16].

The eSRT is defined as the lowest level of electrical stimulation to elicit a change (0.02) in static admittance in response to electrical stimulation via CI [17]. The eSRT is similar to acoustic reflex thresholds whereby stapedius muscle contraction (static admittance) is monitored in response to acoustic stimuli delivered through the CI [18]. The eSRT may be obtained intraoperatively via direct visual observation, standard immittance measures or electromyographic measurement, or postoperatively via standard immittance measures [11, 19]. Importantly, intraoperatively obtained eSRTs are not correlated with postoperatively obtained eSRTs [20], although suggested to be due to surgical factors or differences in measurement approach, the precise reason for this mismatch is currently unclear. However, eSRTs have previously led to overestimation of maximum comfortable stimulation levels, potentially leading to overstimulation [2]. In addition, up to 30% of patients tested for eSRT show no measurable response, limiting the possible clinical use [21].

Previous and current MED-EL fitting software (MAESTRO 9.05) offered automatic eCAP measurement known as Auditory Nerve Response Telemetry (ART). However, the correlation between ART and M levels is much lesser than eSRT and M levels [22]. ART is elicited at much lower levels than where M levels should be mapped in the electrical dynamic range; this is due to the eCAP measured by ART likely reflecting the initial level of auditory perception. In contrast, eSRT is typically elicited at a very high stimulation level, much closer to behavioural M levels. Further, eSRT offers several clinical advantages. Firstly, electrode threshold recording of eSRT (mean 35 seconds) is significantly faster than eCAP (mean 120 seconds) [9]. Secondly, it requires minimal cooperation and is unaffected by arousal, which is important when working with difficult-to-condition populations [23]. eSRT measurements are performed using the same software for CI fitting procedures. Thus, parameter settings (pulse duration, repetition rate) are identical to those used to obtain psychophysical behavioural levels [24]. In addition, eSRT has been shown to remain stable over time [11, 25].

A recent systematic review found approximately 41–44% of adults with dementia could not successfully complete pure-tone audiometry (three-frequency pure-tone average bilaterally [26]. Given CI mapping (M-Level) requires additional judgement about an acoustic signal beyond its basic presence or absence [27], subjective loudness judgement for CI mapping procedures may be considered inappropriate for these CI subgroups. This study aimed to investigate whether eSRTs (i) can be reliably elicited in adult CI recipients to generate data for CI programming procedures, (ii) provides equivalent speech outcomes to the conventional subjective behavioural method, (iii) provides equivalent sound quality perception to the conventional subjective behavioural method, and (iv) whether cognitive state affects speech reception performance with either programming method.

## Materials and methods

### Ethics

The University of Western Australia Ethics committee provided ethics approval for this study (RA/4/1/7369). All procedures adhered to this approval. Participants completed a consent form before taking part in the study.

### Participants

Unilateral CI recipients (aged 18–85 years, mean of 67.0) with at least six months CI experience and three months stability (MAP and impedance) were recruited from the Ear Science Implant Centre, Perth. For better comparison, only participants with a MED-EL multichannel CI (MED-EL Manufacturers, Innsbruck, Austria) were invited. Participants with pre-lingual deafness, known speech impediments, or contralateral severe middle-ear dysfunction were

excluded. Qualified participants were randomly divided into two age-matched groups (Group A and Group B) before completing measures in the given order.

## Cognitive assessment

Cognitive assessment was performed using the validated Montreal Cognitive Assessment—Hearing Impaired (MOCA-HI), a variation of the MoCA test adapted for users with hearing impairment and described in detail previously [28]. This variation uses a timed Microsoft PowerPoint presentation (Microsoft Corp., Redmond, WA), with all verbal instructions replaced by visual instructions [29]. The clinical determination of cognitive impairment is made with a threshold for cut-off of $\leq 23$, with all those scoring $> 24$ considered to have normal cognitive function [23].

## Study conduct

Participants were required to attend three sessions within a 4-week period. During session one, a demographic questionnaire was completed, the MoCA-HI cognitive assessment was performed and immediately followed by the generation of two MAPs, one objective eSRT-based MAP and subjective psychophysical-based MAP. Both MAPs were de-identified and blinded from participant and researcher responsible for speech assessments. Participants in Group A were fit with their subjective psychophysical-based MAP. Participants in Group B were fit with their Objective eSRT-based MAP. Both groups trialled given MAP for a two-week period [16, 30]. Following the two-week trial, all participants attended session two, whereby participants completed the Hearing Implant Sound Quality Index (HISQUI) and speech test battery using trialled MAP.

Participants were given alternate MAP to trial for an additional two-week period (Group A: Psychophysical-based MAP, Group B: eSRT-based MAP). During session three, participants completed the HISQUI and speech test battery using the second trial MAP. Following study completion, participants had their original MAP returned.

## Outcome measures

**eSRT-based M-level & MAP.**  Participants were instructed to indicate if the stimulus became uncomfortable. If no response occurred, the stimulus was increased (6%). Once a repeatable reflex was recorded, the stimulus was decreased until no reflex was present (Fig 1). The eSRT was taken as the lowest stimulus level that produced a definitive, repeatable deflection in the baseline recording of at least 0.02ml following, synchronous with the stimulus presentation [17]. Electrodes with no recordable eSRT had M-Levels interpolated from adjacent electrodes. The M-Level was set as eSRT level in software. Global loudness adjustments were performed if required. This allowed volume modifications whilst maintaining inter-electrode eSRT-based relationships. All deviations from eSRT were recorded. Patients with no reliable eSRT responses were excluded from further analysis.

**Psychophysical-based M-level & MAP.**  Participants rated stimulus presentation on a six-step loudness scale. Electrical stimulation levels were systematically increased relating to 15% of the dynamic range until the participant indicated that sound was 'very loud, but not uncomfortable'. Step increases were reduced to 3% and increased systematically until the participant reached 'uncomfortably loud' (Fig 1). Stimulation was reduced until the 'very loud but not uncomfortable' level (M-Level) was reached. This process was repeated until the participant exhibited consistent M-Level judgement before moving to the next electrode. Global loudness adjustments were performed if required.

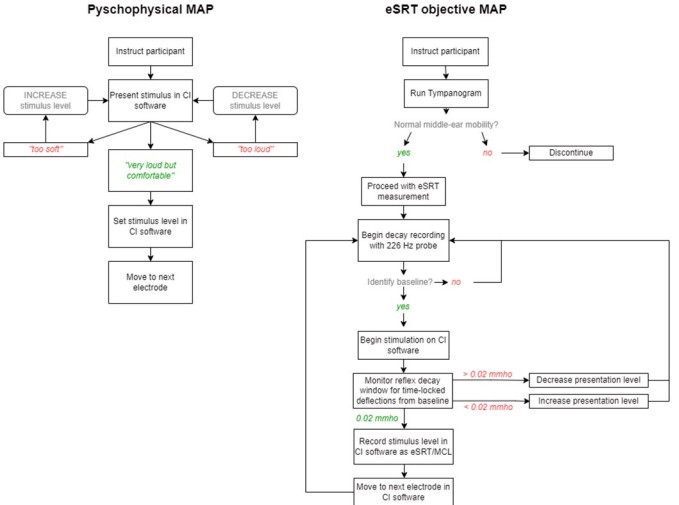

**Fig 1. Flowcharts of psychophysically-determined and electrically evoked Stapedial Reflex Threshold (eSRT) mapping procedures.** CI refers to Cochlear Implant. Mmho refers to units of air pressure.

**Speech assessment measures.** A speech test battery assessed functional MAP outcomes [30 minutes]. Three speech assessments were completed in a randomised order: Speech-in-quiet City University New York (CUNY) Sentence Test [31], consonant-nucleus-consonant (CNC) open-set word test [32], Bamford-Kowal-Bench Speech-in-noise (BKB-SIN) sentence test [33]. Speech stimuli were presented at 65dB SPL (average conversational level) zero-degrees azimuth at a distance of one meter in a soundproof booth unless otherwise stated. Appropriate speech masking was presented to the contralateral ear.

**Hearing implant sound quality index.** Participants completed the 29-question HISQUI questionnaire [34] to assess perceived CI sound quality for both MAPs. Responses were rated on a seven-point Likert scale and presented as a percentage score. A total score indicates sound quality in personal everyday listening situations. Higher scores were indicative of better sound quality percept.

## Statistical analysis

Data were statistically analysed using SPSS v25. The mean difference between eSRT-based outcomes and psychophysical-based outcomes were investigated with paired t-test analysis. Pearson correlation was used to examine the association of M-Levels between subjective and objective MAPs. Pearson correlation was used to investigate any association between participant demographic data and eSRT level. Kendall's Tau-B was used to investigate associations between MoCA-HI scores and speech reception results. P Values <0.05 were considered statistically significant.

## Results

### Participants

Twenty-seven post-lingually hearing impaired MED-EL CI recipients (M = 67.4 years ± 12.7 years) were recruited. A measurable eSRT was successfully obtained in 23 of the participants (85%), those without measurable eSRT were excluded from further analysis. All participants (100%) had tympanograms within the normal range; therefore, no participants were excluded based on clinically significant atypical middle-ear function (Table 1). Six participants showed

**Table 1. Participant demographic data.**

| ID | Sex | Age | HL Aetiology | Years of HL | BE3PTA | CI Type [Sound Processor] | CI Use (Months) |
|---|---|---|---|---|---|---|---|
| 1 | M | 76 | Meniere's Disease | 5 | 23 | SYNCHRONY FLEX24 [SONNET] | 48 |
| 2 | M | 40 | Unknown | 30 | 71 | CONCERTO FLEX28 [RONDO] | 66 |
| 3 | F | 58 | Acoustic Schwannoma | 3 | 3 | SYNCHRONY FLEX28 [SONNET] | 19 |
| 4 | F | 70 | Unknown | 60 | 71 | SONATAti100 FLEX24 [OPUS 2] | 107 |
| 5 | F | 71 | Unknown | 11 | 93 | SYNCHRONY FLEX24 [SONNET] | 48 |
| 6 | F | 53 | Labyrinthitis | 1 | 10 | SYNCHRONY FLEX28 [SONNET] | 36 |
| 7 | M | 76 | Ototoxicity/ Noise | 20 | 53 | SYNCHRONY FLEX28 [SONNET] | 22 |
| 8 | F | 52 | Viral Meningitis | 24 | 30 | SYNCHRONY FLEX24 [RONDO] | 11 |
| 9 | F | 66 | Meniere's Disease | 9 | 20 | SYNCHRONY FLEX28 [SONNET] | 51 |
| 10 | F | 48 | Mumps | 40 | 16 | CONCERTO FLEX28 [SONNET] | 57 |
| 11 | F | 41 | Acoustic Schwannoma | 20 | 42 | SYNCHRONY FLEX24 [SONNET] | 21 |
| 12 | F | 61 | Schwannoma | 6 | 2 | SYNCHRONY FLEX28 [SONNET] | 41 |
| 13 | M | 74 | Noise-Induced HL | 59 | 55 | SYNCHRONY FLEX24 [SONNET 2] | 18 |
| 14 | M | 79 | Meniere's Disease | 5 | 23 | SYNCHRONY FLEX24 [SONNET 2] | 72 |
| 15 | F | 83 | Unknown | - | 72 | SYNCHRONY FLEX28 [SONNET] | 63 |
| 16 | F | 65 | Acoustic Neuroma | 4 | 7 | SYNCHRONY FLEX28 [SONNET] | 39 |
| 17 | F | 73 | Unknown | - | 87 | SONATA EAS FLEX [SONNET 2] | 136 |
| 18 | M | 80 | Noise-Induced HL | 63 | 75 | SYNCHRONY FLEX SOFT [SONNET] | 31 |
| 19 | M | 80 | Acoustic Neuroma | 25 | 62 | SYNCHRONY FLEX28 [SONNET] | 36 |
| 20 | M | 73 | Noise-Induced HL | 33 | 55 | SYNCHRONY FLEX28 [SONNET] | 32 |
| 21 | F | 73 | Meniere's Disease | 13 | 37 | SYNCHRONY [SONNET] | 43 |
| 22 | M | 81 | Noise-Induced HL | 55 | 52 | SYNCHRONY FLEX28 [SONNET] | 66 |
| 23 | F | 67 | Usher's Syndrome | 54 | 60 | SYNCHRONY FLEX24 [SONNET 2] | 21 |

*Note.* ID = Participant Identification Number, M = Male, F = Female, HL = Hearing Loss, HA = Hearing-aid, BE3PTA = Three Frequency (500, 1000, 2000Hz) Pure Tone Average of Better Ear, CI = Cochlear Implant.

MoCA-HI results ≤ 23 indicating cognitive impairment, with the cognitive impairment group significantly older, but otherwise not significantly different from the normal cognition group (Mann-Whitney U = 12.0, $p = .007$ for age, $p$ values of .13, .78, and .13 for sex, months CI usage, and years HL respectively).

## eSRT-based and psychophysical-based M-Levels

Overall charge for patients between eSRT and psychophysical-based M-Levels demonstrated a significant, strong, positive correlation (Fig 2, right panel, $r = 0.89$, $p < .001$).

A paired sample t-test was performed to determine the mean group difference between psychophysical-based and eSRT-based M-Levels at each electrode location (Table 2). Overall, there was no significant difference between the overall MAP charge across the array. Analysis of individual electrodes demonstrated that the behaviourally-determined MAPs had significantly higher M-Levels on electrodes 6–8.

## Speech assessment outcomes

**Speech-in-noise.** Mean speech outcome scores are presented in Fig 3. Speech-in-noise (BKB-SIN) scores represent the mean of two sentence lists using the keyword loose method [35], where keywords are scored without penalty for plural errors. Scores, here represented by the signal-to-noise ratio, ranged from -3 dB (noise higher than speech), to 21.5 dB (speech

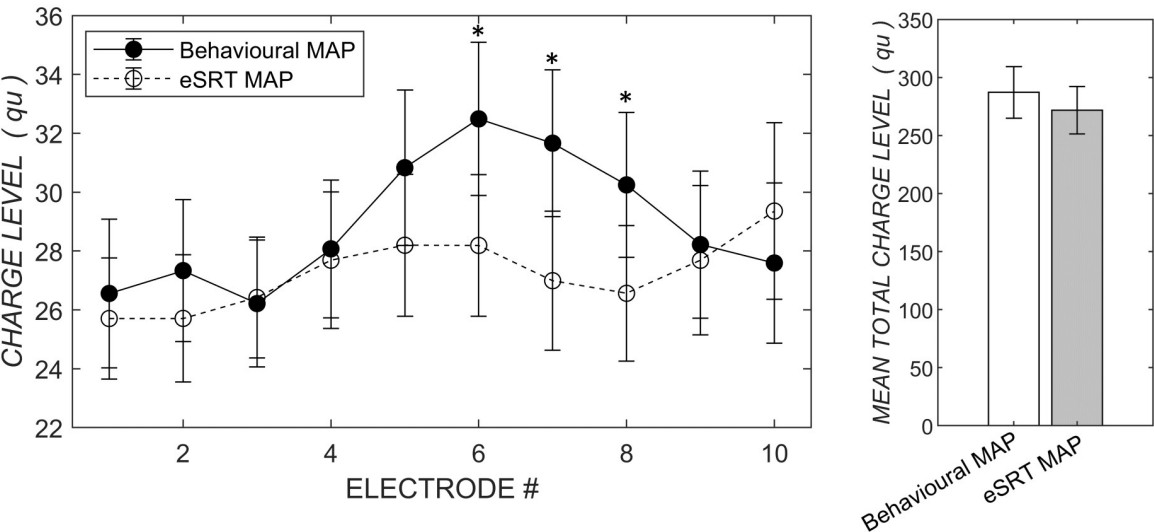

**Fig 2. Left panel, comparison of charge level across the electrode array for behavioural and eSRT-determined MAPs.** Right panel, comparison of mean total charge across the array for the two mapping approaches. * $p < .05$. Error bars for standard error.

louder than noise). The maximum SNR for the BKB-SIN test is 25-dB, a ceiling not reached by any participants in this study.

Patients demonstrated significantly better BKB-SIN scores (lower signal to noise-ratios) with the behaviourally-determined MAP in comparison with the objectively determined MAP (means of 6.24 dB for the behaviourally determined MAP, 8.2 dB for the objectively determined MAP, paired t-test, t = -2.26, $p = .034$). Grouping patients into normal cognition and cognitively impaired groups showed no significant difference in these scores (t = -1.66, $p > .11$ for normal cognition, t = -1.60, $p > .17$ for cognitive impaired group).

Shown in Fig 4 and Table 3, cognitive state demonstrated a significant, negative correlation between MoCA-HI score and BKB-SIN results for both behaviourally and objectively determined MAPs, with the objectively determined test slightly more affected by MoCA-HI result. To assess whether MoCA-HI score affects the *difference* between BKB-SIN score when tested

**Table 2. Paired sample T-Test of psychophysical-based and eSRT-based M-Levels.**

|  | Paired Differences | | 95% Confidence Interval of the Difference | | | | |
|---|---|---|---|---|---|---|---|
|  | **Mean** | **Std.Dev** | **Lower** | **Upper** | **t** | **df** | **Sig. (2-tailed)** |
| Electrode 1 | 0.85 | 1.64 | -2.55 | 4.26 | 0.519 | 22 | .609 |
| Electrode 2 | 1.62 | 1.58 | -1.66 | 4.91 | 1.025 | 22 | .316 |
| Electrode 3 | -0.21 | 1.36 | -3.00 | 2.59 | -0.152 | 22 | .881 |
| Electrode 4 | 0.39 | 1.38 | -2.47 | 3.25 | 0.280 | 22 | .782 |
| Electrode 5 | 2.63 | 1.40 | -0.27 | 5.53 | 1.884 | 22 | .073 |
| Electrode 6 | 4.30 | 1.19 | 1.83 | 6.76 | 3.617 | 22 | **.002** |
| Electrode 7 | 4.67 | 1.28 | 2.02 | 7.32 | 3.663 | 21 | **.001** |
| Electrode 8 | 3.68 | 1.05 | 1.50 | 5.87 | 3.510 | 21 | **.002** |
| Electrode 9 | 0.53 | 1.48 | -2.55 | 3.61 | 0.360 | 21 | .723 |
| Electrode 10 | -1.76 | 1.36 | -4.62 | 1.09 | -1.294 | 19 | .211 |
| Overall MAP | 15.37 | 9.94 | -5.25 | 35.98 | 1.546 | 22 | .136 |

*Note.* Asterisk (*) represents a statistical difference (P < 0.05). Std. Dev = Standard Deviation, t = T Value, df = Degrees of Freedom, Sig = Significant Difference.

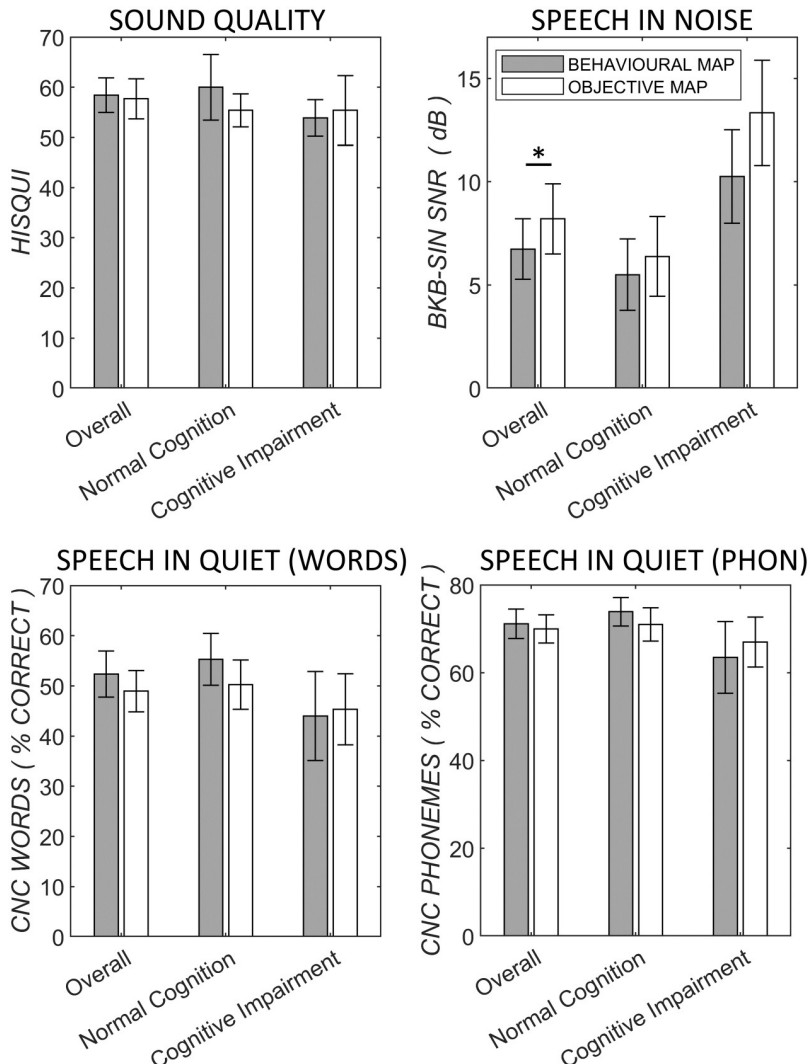

**Fig 3. Sound quality and speech reception scores for all patients and patients grouped by cognitive state.** Results for both the behaviourally determined MAP (grey bar) and the eSRT-based objectively determined MAP (white bar) are shown. * for $p < 0.05$.

using the behavioural versus objectively determined MAPs, a correlation was made between MoCA-HI score and the difference between BKB-SIN scores. There was no significant correlation found between BKB-SIN score difference and MoCA-HI score (Kendall's Tau B = -0.257, $p = .105$).

**Speech recognition in quiet.** Mean CNC word-in-quiet scores achieved with the eSRT-based MAP were 49.0 ± 20.2% (Range: 18%–92%) and thus slightly lower than those with the behavioural map at 52.3 ± 22.5% (Range: 12%–100%). However, a paired t-test analysis showed no significant differences in scores achieved using eSRT-based or psychophysical-based MAPs (t = 1.17, $p = 0.255$).

Compared with MoCA-HI scores, for the behaviourally-assessed MAP there was a significant, positive correlation between cognitive score and CNC Word scores, such that those without cognitive impairment performed better on this test (Kendall's Tau B = 0.312, $p = .049$). This was not significant for the objectively determined MAP (Kendall's Tau B = .230, $p = .145$).

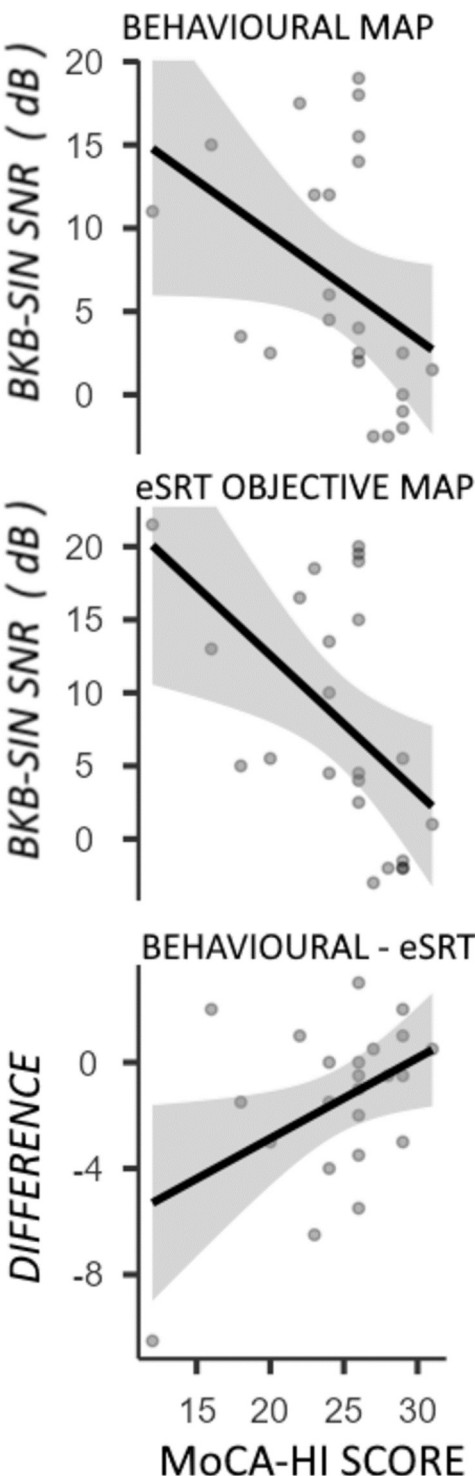

**Fig 4. Correlation between MoCA-HI score and speech-in-noise results using the behaviourally-determined map (top panel), the eSRT objectively-determined MAP (middle panel), and as a difference between these two MAPs (bottom panel).**

**Table 3.  Correlation matrix for MoCA-HI and BKB-SIN scores, separated by cognitive group.**

|  |  | MoCA-HI | Behavioural MAP |
|---|---|---|---|
| Behavioural MAP | Kendall's Tau B | -.383* |  |
|  | *p* value | 0.0015 |  |
| eSRT MAP | Kendall's Tau B | -0.421** | 0.742*** |
|  | *p* value | 0.008 | < .001 |

* *p* < .05,

** *p* < .01,

*** *p* < .001

### Sound quality outcomes

A paired t-test analysis showed no statistical difference between mean overall sound quality scores measured with the HISQUI test between eSRT-based (M: 57.7 ± 16.0%) and psycho-physical-based methods (M: 58.4 ± 16.9%). There was no correlation between sound quality outcomes and MoCA-HI score for either behaviourally or objectively-determined MAPs (For behaviourally MAP, Kendall's Tau-B = 0.122, *p* = .43, for objective MAP, Kendall's Tau-B = -.025, *p* = .87).

### Participant demographic correlations

Participants age was negatively correlated with MoCA-HI score (Kendall's Tau-B = 0.595, *p* < .001), and objectively determined BKB-SIN score (Kendall's Tau-B = .433, *p* = .039). Months of implant use was not correlated with any eSRT or behaviourally determined MAP outcomes.

### Discussion

Cochlear implantation is an effective treatment for moderate-to-profound hearing impairment, and as such implant candidacy has expanded in recent years to include a greater number of older adults [36]. This cohort, as well as the paediatric cohort of implant recipients which continue to be a focus for implantation [37], present challenges in the typical approach to cochlear implant programming, which relies heavily on behavioural tests for setting thresh-old and maximum comfort levels for implantation. In particular, declining cognitive function in the older patient cohort can present substantial challenges for the successful completion of the device mapping procedure [27, 38]. Incorrect assignment of these levels may result in poor implant performance, in particular inappropriately high maximum comfort levels can lead to painful stimulation [3]. An objective assessment of maximum comfort level would bypass these challenges, and enable the rapid mapping of patients otherwise unable to complete beha-vioural tests.

As demonstrated in previous studies, we report here a strong correlation between the maximum comfort levels in a cochlear implant MAP (here, M-Levels) when determined behaviourally compared with those determined with an objective, eSRT based approach. While the global charge rates of the two approaches were not significantly different, the behavioural MAPs featured higher M-Levels along the mid-basal electrodes 6–8 compared with the eSRT MAPs. The cause of this mid-basal enhancement is unclear, and is contrast with prior studies reporting similar, higher behavioural thresholds (Asal *et al.*, 2016, 26 MED-EL cochlear implant recipients), or very similar thresholds (Çiprut and Adıgül, 2020, 46 MED-EL and Cochlear Ltd implant recipients). Early work on eSRT and behaviourally-

determined maximum comfort levels suggested that while these are typically close together, eSRT thresholds can be either over- or under-estimated for some [3]. Some of this variability is likely to be caused by methodological and demographic approaches, with duration of implant use, duration and onset of deafness, and age of implantation identified as contributing factors [39–41].

A potential explanation for higher behavioural thresholds is the identification that behavioural MAP thresholds tend to increase over the implant users experience with the device after implantation, only stabilizing over 24 months of use [39], which is not the case for the highly-stable eSRT-based M-levels [13, 42]. Regardless of mechanism, electrodes 6–8 in a typical MED-EL device using the logarithmic fine structure processing strategy represents an approximate frequency region of 1–3 kHz [43]. This is a key frequency range for speech discrimination and understanding [44], and this may explain the primary effect of the difference in behaviourally determined and objectively determined MAP level, a significant reduction in speech-in-noise intelligibility in the eSRT-based MAP. The "bell shape" in the behaviourally-determined MAP has been widely reported in prior literature, and has been suggested to yield more high-frequency information important for speech intelligibility [30].

The novel finding in the present work is in the comparison of MAP and speech outcomes with cognitive state, using a variant of the MoCA test appropriate for the hearing impaired (MoCA-HI). In particular, MoCA-HI scores correlated negatively with BKB-SIN scores [45, 46], such that those with cognitive impairment perform worse in the speech-in-noise test, indicative of central auditory processing impairment. Programming approach, behavioural or objective, had no effect on this relationship. One concern was that patients with low MoCA-HI scores may not have their M-Levels adequately evaluated using the more-challenging, behavioural approach to cochlear implant mapping. To test this, we assessed whether MoCA-HI score correlated with the *difference* between objective and behavioural BKB-SIN scores, with the hypothesis that objectively determined MAPs would result in higher BKB-SIN scores than behaviourally determined MAPs. No significant correlation was found, suggesting that the relationship between poor BKB-SIN scores and cognitive state is not affected by programming approach. Overall, this finding supports the use of the objective eSRT approach in individuals with normal cognitive function or MCI. Further studies are required to assess whether eSRT can be used in individuals with dementia.

This appears to be the first study investigating the sound quality perception of eSRT-based MAPs in adult CI recipients. In this study, MAP sound quality was measured using the HISQUI, which evaluated how well everyday listening tasks were completed with the CI after each two-week trial. Statistically similar mean overall sound quality scores were reported with both methods. As a validated tool of self-perceived CI auditory benefit [34, 47], the HISQUI determined no notable differences in common listening challenges in CI recipients between mapping methods.

The majority of study participants verbally reported a personal preference of the eSRT-based MAP as it provided improved clarity. A potential explanation for the discrepancy in participant preference and sound quality scores may be that the sound quality outcome measure was not sensitive enough to discern subtle clarity differences between MAPs. For example, Question 7 asks participants to rate how well they understand the voices of unfamiliar people over the phone on a 7-point Likert scale from 'Always' to 'Never'. For CI recipients, phone-related tasks are typically difficult [48, 49], thus 'Rarely' or 'Occasionally' were commonly marked in both scenarios. This may not fairly represent MAP clarity differences, rather just the difficulty of the task itself. This finding encourages the addition of a qualitative measure of sound quality compared to help reflect this finding in future studies. An important consideration in sound quality self-assessment may be that an individual's perception of CI benefit

may not correspond with more objective outcomes. Therefore, caution must be made when extrapolating from results.

A potential limitation in the present work is the limited trial duration for each MAP, over a period of 2 weeks. This may not have been sufficient time for the participants to acclimatise to each MAP, which would obscure the potential benefits or drawbacks of the objectively-determined approach. While the optimal trial duration is unclear, prior work has identified experience-related changes to the brain over 8- to 12-weeks following CI programming procedures, a substantially longer time period of acclimatization [50].

A second limitation is the relatively small number of patients with ≤ 23 scores on the MoCA-HI test. This results in a limited population to test the effect of cognitive impairment on programming approaches, and although differences between MoCA-HI score and BKB-SIN result were clear in the present work, suggesting sufficient variability in the measure, an effect of mild-cognitive impairment on programming outcomes may be obscured by the limited patient number. This is further compounded by the wide variability in patient demographics, including age, years of hearing loss and BE3PTA. This was a stated goal of the study, as this variability was chosen as a possible way of capturing differences between MAP outcomes that may have been present only in some subpopulations. However, this raises the possibility of introducing too much variability to enable the detection of real differences in the population.

## Author Contributions

**Conceptualization:** Jessica M. Yiannos, Barbara Gell, Dona M. P. Jayakody.

**Data curation:** Jessica M. Yiannos, Christofer W. Bester, Christopher Zhao.

**Formal analysis:** Christofer W. Bester, Christopher Zhao, Dona M. P. Jayakody.

**Investigation:** Jessica M. Yiannos, Christopher Zhao, Barbara Gell, Dona M. P. Jayakody.

**Methodology:** Jessica M. Yiannos, Dona M. P. Jayakody.

**Project administration:** Dona M. P. Jayakody.

**Resources:** Jessica M. Yiannos, Barbara Gell.

**Supervision:** Dona M. P. Jayakody.

**Writing – original draft:** Christofer W. Bester.

**Writing – review & editing:** Jessica M. Yiannos, Christofer W. Bester, Dona M. P. Jayakody.

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
