## [Decision Letter · Decision Letter 0]

7 Mar 2023

PONE-D-23-00636Speech-in-noise performance in objectively determined cochlear implant maps: evaluating the effect of cognitive state.PLOS ONE

Dear Dr. Bester,

Thank you for submitting your manuscript to PLOS ONE. After careful consideration, we feel that it has merit but does not fully meet PLOS ONE’s publication criteria as it currently stands. Therefore, we invite you to submit a revised version of the manuscript that addresses the points raised during the review process.

We look forward to receiving your revised manuscript.

Kind regards,

Prashanth Prabhu

Academic Editor

PLOS ONE

Journal Requirements:

"It is disclosed that one of the authors is a clinical specialist working for MED-EL Australasia. "

Reviewers' comments:

Reviewer's Responses to Questions

**Comments to the Author**

1. Is the manuscript technically sound, and do the data support the conclusions?

Reviewer #1: Partly

2. Has the statistical analysis been performed appropriately and rigorously? 

Reviewer #1: Yes

3. Have the authors made all data underlying the findings in their manuscript fully available?

Reviewer #1: Yes

4. Is the manuscript presented in an intelligible fashion and written in standard English?

Reviewer #1: Yes

5. Review Comments to the Author

Reviewer #1: The title of the present study is “Speech-in-noise performance in objectively determined cochlear implant maps: evaluating the effect of cognitive state” in which authors made an attempt to compare speech reception outcomes between subjective and eSRT objectively determined CI maps for Adult MED-EL recipients. In addition, the effect of cognition skills was also looked into on cochlear implantees. The present study is having clinical relevance for the clinicians and provides evidences for in terms of best method to perform MAP i.e. while performing mapping for the cochlear implantees of different age groups as well as difficult-to test Populations including individuals with cognitive impairment. The specific comments are mentioned in the PDF files. Manuscript needs revision before evaluating further for consideration to publish in the present form.

6. PLOS authors have the option to publish the peer review history of their article (what does this mean?). If published, this will include your full peer review and any attached files.

Reviewer #1: No

---

## [Author Response · Author response to Decision Letter 0]

24 May 2023

Response to editor:

The documents have been updated for style.

"It is disclosed that one of the authors is a clinical specialist working for MED-EL Australasia. "

Thank you, I can confirm that this does not alter our adherence to PLOS ONE policy, and this has been updated in the cover letter.

Thank you, unfortunately we have not been able to upload the data while abiding by Dryad’s rules around re-identifiable human data, i.e. replication requires more than 3 indirect identifiers. I do not believe we will be able to upload this data in an acceptably anonymized format.

Ok! Thanks for this catch!

Seems ok to us.

Response to reviewer:

In line no. 80-85, it is mentioned that objective mapping procedures include eCAP, eABR and eSRT. Later, the description of eSRT mentioned by the authors. It will be interesting and meaningful for the reader if the test-retest reliability of the eSRT mentioned here based on the literature. In addition, in normal circumstances, the presence and absence of the eSRT in different age groups (Children, adults and older adults) based on literature will be meaningful for the readers. Prior to eSRT, few lines about the eCAP and eABR over eSRT preference by the clinician with the literature support will also be valuable. 

This is a very good point, we have added the details on the eCAP and eABR in the paragraph, detailing the relatively poor correlation between eCAP and eABR thresholds and the behavioural scores we are interested in, and the risk of these thresholds being more than 5 programming units higher than the most which eSRT may be able to improve upon.

In line 93-94, the mentioned statement “intraoperatively obtained eSRT are not correlated with postoperatively obtained eSRT” supported with reason from the existing literature will add value to it.

It is very interesting – the articles on the topic don’t advance concrete theories for the discrepancy! A sentence has been added raising the potential for measurement differences or surgical impact as confounding factors.

In line 102-109, the several clinical advantages of the eSRT are mentioned. It will be good if the limitation of the eSRT too mentioned if any.

Added information that the eSRT, as classically measured, overestimates maximum comfort levels (MCLs) which may lead to overstimulation and thus must be used with caution. More concerningly, up to 30% of patients show no measurable reflexes, limiting the possible clinical use. 

Line 133: Since authors targeted 18-85 years of unilateral CI recipients for the study, one paragraph about the possible factors influencing the CI mapping including outcome might help the readers.

A sentence referencing the possible factors influencing CI map, and how this map affects CI outcomes have been added to the first paragraph of the introduction.

In line 130, the mean age of the participants can be added. 

This has been added.

Since participants are adults and older adults, any screening test for auditory processing disorders (APD) could have been ruled out. 

Unfortunately, no APD testing was carried out in this population.

In line 173, authors should mention the no. of participants excluded due to no reliable eSRT responses. 

The number of patients without a reliable eSRT is included in the results (N = 4), this line has been removed as misleading in the methods.

In line 176, the testing time for conducting psychophysical-based M-level & MAP can be added at the end of the paragraph.

Unfortunately, testing time was not recorded during these sessions.

In line 192, authors should mention that for how many participants speech masking was used while administrating speech assessment measures.

Unfortunately, this information was not recorded during these sessions.

In line 196-199, For HISQUI questionnaire maximum score and the time taken to perform the test should be mentioned. Further, it should also be mentioned.

The HISQUI score is reported as a percentage out of a possible maximum of 133, we have clarified this in the methods and the results.

Line 211: 27 can be written in words as "Twenty-seven"

Done!

Table 1: How authors justify the large variation in the Implantation age, years of hearing loss, BE3PTA across participants. Any insight on these factors in discussion will be interesting for the reader. 

This wide variability was a potential strength of the study, as it gives the best chance of catching a difference between the objective and behaviourally determined MAPs – i.e. if older patients, or patients with lower BE3PTA scores, had larger discrepancies between MAP approaches, we may have caught them using this approach. Of course, this adds the possibility that we are adding too much variability to catch real changes. A paragraph on potential limitations has been added in the discussion.

Line 229: replace with "and"

Done!

Line 236: Authors should made an attempt to explain the possible reason for the significantly higher M levels in mid electrode position. 

This is a difficult point, as there are many contributing variables that would have to be assessed individually to assess the possible mechanisms. We have outlined the difficulty of reasoning this difference, and added further comparison with other publications. A possible addition to the discussion could be to investigate these changes with appropriate controls for demographics, insertion dynamics?

Line 257: What could be the limitation comparing only 6 participants with cognitive impairment with rest of the participants with normal cognitive function. I think author do mentioned as one of limitations. 

This is limiting, and has been described in the discussion.

Line 308: In objective assessment, Authors can throw insight about how appropriate while measuring eSRT in comparison to eCAP and eABR.

As per a previous reviewer comment, this has been addressed in the introduction, and would repetitive if included in the discussion as well.

Line 343: As authors pointed out that auditory processing impairment could be the reason for poor SPIN test performance. It would be more meaningful if author can elaborate further to substantiate the reason mentioned for poor SPIN score. 

As the difference in SPIN score is between the two MAP types, i.e. a behaviourally determined MAP provided significantly better SPIN scores than the objectively determined MAP, it is unclear how auditory processing impairment could contribute to the discrepancy. We have included a possible explanation for this difference in lines 341-349, that the increased M-level in the objectively-determined MAP along the mid-basal electrodes is in a prime frequency range for fine-structure processing in the speech-discrimination and understanding frequency range.

---

## [Editor Report · Decision Letter 1]

30 May 2023

Speech-in-noise performance in objectively determined cochlear implant maps, including the effect of cognitive state

PONE-D-23-00636R1

Dear Dr. Jayakody,

We’re pleased to inform you that your manuscript has been judged scientifically suitable for publication and will be formally accepted for publication once it meets all outstanding technical requirements.

Kind regards,

Prashanth Prabhu

Academic Editor

PLOS ONE
---

## [Editor Report · Acceptance letter]

5 Jun 2023

PONE-D-23-00636R1 

Speech-in-noise performance in objectively determined cochlear implant maps, including the effect of cognitive state. 

Dear Dr. Jayakody:

I'm pleased to inform you that your manuscript has been deemed suitable for publication in PLOS ONE. Congratulations! Your manuscript is now with our production department. 

Kind regards, 

on behalf of

Dr. Prashanth Prabhu 

Academic Editor

PLOS ONE